# Cross-Sectional Survey of Mental Health Risk Factors and Comparison of the *Monoamine oxidase A* Gene DNA Methylation Level in Different Mental Health Conditions among Oilfield Workers in Xinjiang, China

**DOI:** 10.3390/ijerph17010149

**Published:** 2019-12-24

**Authors:** Ting Jiang, Xue Li, Li Ning, Jiwen Liu

**Affiliations:** Department of Public Health, Xinjiang Medical University, Urumqi 830011, China; jt5583@126.com (T.J.); lixue13039438978@163.com (X.L.); nl96979@163.com (L.N.)

**Keywords:** mental health, *MAOA* gene, DNA methylation, health risk factors-evaluation, oil workers

## Abstract

The incidence of psychological problems among occupational groups is becoming increasingly more serious, and adverse psychological conditions will seriously affect the working ability of occupational groups and harm the health of their bodies. This study adopted a multi-stage stratified cluster sampling method to conduct a cross-sectional survey on the mental health of 3631 oil workers in Karamay, Xinjiang from March 2017 to June 2018. The mental health status of oil workers was evaluated using the Symptom Checklist-90, and mental health risk factors were evaluated. The correlation between the monoamine oxidase A (*MAOA*) gene and mental health was analyzed, and the DNA methylation level of the *MAOA* gene was compared between the normal group and the abnormal group. The results show the incidence of mental health problems among oil workers according to differences in age, nationality, type of work, length of service, professional title, shift work, and marital status. The evaluation of mental health risk factors revealed that shift work, occupational stress, and high payment/low return affect mental health. The somatization scores of different genotypes of rs6323 in the *MAOA* gene were statistically significant (*p* < 0.05), suggesting that the somatization scores of different genotypes of rs6323 were different. According to the average rank, the TT genotype group had the highest score, followed by the GT genotype group, and the GG genotype group had the lowest score. The level of DNA methylation in the abnormal group was lower than that in the normal group (*p* < 0.05). The results suggested that occupational mental health can be enhanced by improving shift work, reducing stress, and balancing effort and reward. This preliminary investigation suggests that methylation status can affect mental health, indicating that methylation level may be a predictor of mental health status.

## 1. Introduction

In modern society, fierce job competition and tense interpersonal relationships can easily cause mental health problems among professional people. Mental health refers to a state of psychological functioning in which an individual has no mental illness or abnormality, is well adapted to society, has a perfect personality and ability, and can fully realize their potential [1]. The physical and mental health of the population and the ability to be socially well-adjusted can directly affect the survival and development of enterprises as well as national social stability [2]. About 450 million people worldwide are reported to suffer from mental illness, and at least one in four families suffers from psychological and behavioral disorders. Approximately 25% of the UK population experiences mental disorders every year [3]. Annual incidence of mental disorders in Ghana is 13% [4]. The prevalence of mental disorders in China is 12% [5], while the lifetime prevalence of mental disorders in West Africa is 12.1% [6]. According to research data, the economic burden of mental illness was about 2.5 trillion US dollars in 2010, and it is estimated that the cost will increase to 6 trillion US dollars by 2030 [7], which accounts for about 3–4% of gross national product (GNP) in developed countries. However, most of the social burden of mental health problems is not related to the cost of treatment, but to the reduction in human capital and productivity [8].

At present, research on the mental health problems of occupational groups mainly include teachers, medical personnel, skilled workers, oil workers, police, etc. Studies have found that long-term excessive pressure or harsh working environments aggravate the occurrence of mental illness [9,10,11,12,13]. Recent research on the influencing factors of mental health mainly analyze the relationship between gender, age, education, marriage, physical health, and other population characteristics and mental health (e.g., Yu Shanfa [14], Yu Xiaoxia [15], Qiu Xingyuan et al. [16]). Some scholars also found that work time, work status, and work arrangements were correlated with psychological problems. Excessive working hours, frequent shift work, and high workloads lead to a series of negative emotions [17,18,19]. The causes of psychological problems are numerous and are often a combination of multiple factors. Therefore, by evaluating the health risk factor, some scholars have studied the probability of disease occurrence in the environment by examining risk factors, as well as the change in the disease occurrence risk factor in response to changes in unhealthy behaviors and by eliminating or reducing risk factors [20].

Health risk appraisal (HRA) is a technical method that is employed to study the quantitative dependence and regularity between risk factors and the incidence of chronic diseases or deaths [21]. HRA mainly examines the impact of various risk factors in terms of the occurrence and development of diseases in the production environment, lifestyles, and medical and health services. By taking effective measures, risk factors that can be changed by intervention, such as workload (e.g., working hours, complexity, shift work), interpersonal relationships (superiors and subordinates, colleagues), and poor working environments (noise, dust, chemical poisons), have been proven to reduce the risk of disease. Studies [22,23] have confirmed that the evaluation of health risk factors plays a positive role in disease prevention.

With the rise of molecular genetics, more and more researchers are beginning to pay attention to the effect of gene polymorphism on mental health. It was found that monoamine oxidase A (MAOA) mainly acts on the limbic system which is associated with higher cognitive functions such as emotional regulation. The gene, located on the X chromosome, degrades neurotransmitters such as serotonin, dopamine, and catecholamines in the human brain and affects the psychological state of the individual by regulating the level of neurotransmitters in the brain. Yu et al. [24] proved that either low activity or lack of MAOA contributed to an ineffective degradation of excessive serotonin, which was the direct cause of depression. A study explored the relationship between the *MAOA* gene and alcoholics with depression and anxiety by examining the polymorphism of the promoter region of the *MAOA* gene in the blood of German male and female alcoholics, and found that the *MAOA* gene was closely related to the incidence of anxiety and depression among alcoholics [25]. In recent years, epigenetic research has made great advancements. The researchers believe that epigenetic modifications provide a means of explaining the influence of environmental factors on gene function. In addition, epigenetics can also offer a theoretical basis by which to explain the inconsistencies of a large number of studies on the association between mental health problems, such as the different transcription efficiency caused by different epigenetic modification states of gene promoters [26]. DNA methylation is one of the more thoroughly studied mechanisms of epigenetic modification. It is a heritable and precise form of gene regulation provided by eukaryotes with cytosine methylation to assist gene silencing [27]. Methylation of transcription factor binding sites does not significantly affect the specific binding of transcription factors, while methylation of CpG sites in adjacent sequences can significantly reduce the binding activity of transcription factors [28]. The *MAOA* gene has two CpG islands, and the differential methylation status of CpG islands can dynamically affect gene expression and the synergistic effect of other potential transcription factors, resulting in different pathogenicity [29,30].

The onset of psychological problems is influenced by genes, environment and epigenetics. However, most of the current studies focus on the polymorphism of the *MAOA* gene variable number tandem repeat (VNTR), and the majority of studies are related to behavior, such as aggressive behavior and criminal behavior [31,32], and they lack consistent results. Therefore, this study selected the *MAOA* rs6323 locus and *MAOA* rs 1137070 locus to study the genes implicated in psychological problems and analyzed the correlation between both loci and psychological problems. In addition, the DNA methylation level of the *MAOA* gene in the mental health group and the abnormal group was compared, to explore the relationship between DNA methylation mechanism and mental health.

## 2. Materials and Methods

### 2.1. Participants

This study was conducted in the Occupational Health Examination Department of the Central Hospital of Karamay, Xinjiang. The questionnaire was part of the occupational health examination of oil workers, and the survey was conducted between March 2017 and June 2018. The participants in this study included employees from the China National Petroleum Corporation (CNPC), Xinjiang petroleum administration in Karamay, Xinjiang. The administration has 25 subordinate units and about 150,000 employees and carries out all work related to the oil industry. Using the three-stage stratified sampling method, four operating areas, four production plants, and six exploration and development companies were selected according to the standard industrial classification of CNPC. According to the size of the company, a large company (>400 workers) and a small company (<400 workers) were randomly chosen from the selected business area. For the business area, 600 employees from large companies and 300 employees from small companies were randomly selected. A large company (>1000 workers) and a small company (<1000 workers) were selected from the production plant. For the production plant, 1500 employees in large companies and 800 employees in small companies were randomly selected. One large company (>200 workers) and two small companies (<200 workers) were selected from the exploration and development company, 300 people were randomly selected from the large company, and 150 people were selected from each small company. Following communication and negotiation with the hospital, the number and information related to each company’s physical examination personnel were obtained before the physical examination. The number of people who took the physical examination was recorded (starting from 1...) and the number was input into SPSS for Windows, Version 22.0. A random sample of cases was used to extract the required samples, and the subjects were determined according to the physical examination list. The inclusion criteria included the following: Petroleum workers (aged between 18 and 60 years who had worked for more than 1 year) received 3800 questionnaires, and finally, 3631 valid questionnaires were retrieved, with an effective recovery rate of 95%. The participants were measured by the general index, and 10–20% of those who participated in the questionnaire were randomly selected as experimental research objects. Therefore, a total of 726 participants were randomly selected as research objects of molecular biology. After excluding the samples for which blood samples were not collected and those unqualified for DNA extraction, 696 samples were finally tested for gene polymorphism. Subsequently, 23 patients in the abnormal group and 23 patients in the normal group were randomly selected for DNA methylation detection of the *MAOA* gene promoter region. The research scheme was approved by the ethics committee of Xinjiang Medical University, and all participants provided their voluntary written informed consent before the investigation.

### 2.2. Measures

#### 2.2.1. Mental Health Status

The Symptom Checklist 90 (SCL-90), compiled by Derogatis in 1975, was adopted [33]. The scale was translated and introduced to China by domestic scholar Wang Zhengyu [34] in 1986, and Wang Jisheng [35], Li Lingjiang [36], and others believed that the scale had good reliability and validity. The scale consisted of 90 rating items including somatization, compulsive symptoms, interpersonal sensitivity, depression, anxiety, hostility, fear, paranoia, psychosis, and another nine factors. Responses were measured using a scale ranging from 1 to 5, such that 1 = none, 2 = mild, 3 = moderate, 4 = severe, and 5 = severe. According to the Chinese norm, the total score and factor score were taken as indicators to evaluate the mental health level (i.e., if the total score ≥160 points; the score of any factor is >2; or the number of positive items is >43, this indicates that there may be a mild or above level of psychological symptoms, that is, there may be mental health problems) [37].

#### 2.2.2. Evaluation of Individual Health Risk Factors

Based on the methods of Yuan Jianping [38,39,40] and other scholars, a risk score table was compiled to form an individual risk factor evaluation model to assess the impact of the different exposure levels of each risk factor on mental health.

Calculated baseline incidence ratio: The baseline incidence ratio is the ratio of the incidence of the individual with the lowest risk factors to the total incidence of the population. According to the calculation formula proposed by Rothman and Keller [41] $\mathrm{Baseline}\;\mathrm{Incidence}\;\mathrm{Rate}=1/\overset{n}{\underset{r=1}{\sum }}RR_{i}\times P_{i}$. $P_{i}$, which reflects the proportion of individuals exposed to a given level of risk factors in the population. $RR_{i}$, the relative risk of exposure to a level of risk factor.

To calculate the combined risk score: Risk Score (RM) = Baseline Incidence Rate × Relative Risk ($RR_{i}$). For diseases with multiple risk factors, the combined risk score is calculated as follows: 1. If the risk score is less than 1, it is necessary to subtract 1, and then all of the values are added. 2. Multiply all of the risk factors if the risk score ≤1. The combined risk score is obtained by adding the values of step 1 and step 2. $\left(P_{z}\right)=\left(P_{1}-1\right)+\left(P_{2}-1\right)+\cdots +\left(P_{n}-1\right)+Q_{1}\times Q_{2}\times \cdots \times Q_{m}$. $P_{i}$: each hazard score ≥1, $Q_{i}$: each hazard score of <1.

The individual evaluation: At present, a risk score of less than 1 indicates a low risk type. The target risk score was much less than the current risk score, which was self-created. If the target risk score was larger than 1 and the difference between the target risk score and the current risk score was small, it indicated a type of risk factor that was difficult to change. A risk score that was equal to 1 was generally dangerous.

#### 2.2.3. Collection of Blood Samples

Blood samples were collected in accordance with the principles of the Helsinki Declaration. Having obtained the participants’ informed consent, 4 mL vacuum blood vessels were used to collect blood samples from 9 a.m. to 11 a.m. without meals or water for the medical personnel. The blood samples collected on the same day were centrifuged at 3500 rpm for 7 min to separate serum and plasma, and stored at −80 °C.

### 2.3. The Experimental Method

#### 2.3.1. Genotyping

Venous blood samples were collected in EDTA-containing tubes from all participants following a 12-h fast. Genomic DNA was purified from the samples using a whole blood genome extraction kit (Tiangen Biotech, Beijing, China) and cryopreserved at −20 °C until use. Tag single nucleotide polymorphisms (SNPs of *MAOA* were derived from a Chinese Han population in the Haplotype Map database (National Center for Biotechnology Information). SNPs rs6323 and rs1137070 were genotyped with the SNaPshot SNP assay using the primers listed in Table 1. Data were analyzed using GeneMapper 4.1 (Applied Biosystems, Foster City, CA, USA). For quality control, 5% of randomly selected samples were genotyped a second time by different researchers, yielding 100% reproducibility.

#### 2.3.2. Methylation Test of MAOA Gene

The *MAOA* gene promoter region was searched through UCSC Genome Brower website, CpG sites were identified according to the criteria (1. The length of CpG island shall not be less than 500 bp; 2. GC content not less than 55%; 3. The observed/expected dinucleotide ratio was ≥0.65). The Methyl Target technique was used to determine the methylation level in the promoter region of MAOA gene: (1) EZ DNA Methylation-Gold Kit was used for sample processing. (2) The Methyl Target technique divides the selected fragment into two regions for detection, the primers are shown in Table 2. (3) Multiple PCR reaction systems of the same sample were mixed. (4) Specific tag sequences were added to the samples. (5) After quantification, high-throughput sequencing was performed on Illumina Hiseq platform ina 2 × 150 bp double-ended sequencing mode.

### 2.4. Quality Control

#### 2.4.1. Quality Control of Field Questionnaire Survey

Each investigator received formal training to clarify the purpose and significance of the investigation in order to ensure the quality and progress of the investigation. Before the survey, the purpose and significance of the survey was fully explained to the respondents. After obtaining the consent of the respondents, a questionnaire was distributed to them. After the completion of the questionnaire, the investigator carried out a comprehensive check of the completed contents. If there was any doubt, the investigator asked for further verification. In the event that there was any error, the investigator made efforts to correct it in a prompt manner.

#### 2.4.2. Laboratory Quality Control

The laboratory operation was carried out in strict accordance with the requirements of the laboratory. An ultraviolet lamp was used before polymerase chain reaction (PCR), and the disposable nozzle, Eppendorf tube, PCR tube, and other consumables were sterilized using an autoclave prior to use. A blind method was adopted for genetic testing, and samples were randomly selected at 10% to check the consistency of the results. When the results of the retest were inconsistent with the initial test, the experiment was repeated to check again.

### 2.5. Statistical Analysis

Epidata Version 3.1 was used to establish the database. SPSS for Windows, Version 22.0 (SPSS Inc., Chicago, IL, USA) was used for data processing and statistical analysis. Measurement data included $\bar{Χ}\pm S$ for descriptive statistics, a univariate analysis of variance (Fisher’s least significance difference (LSD)) test is used for pairwise comparison if the population is different) was employed, and the nonparametric test method was used to analyze the non-normality and homogeneity of variance. Multiple logistic regression was used to analyze the relationship between genotypes and psychological disorders. The significance level was α = 0.05.

## 3. Results

### 3.1. Psychological Abnormalities with Different Demographic Characteristics

The incidence of psychological abnormalities among different ethnic groups, ages, types of work, lengths of service, professional titles, shift work, and marital status was statistically significant (*p* < 0.05). The incidence among those from a minority background (23.7%) was higher than the incidence found among those of Han ethnicity (22.1%). Among participants of different ages, the incidence among those aged 30–45 years old was the highest, followed by the group ≤30 years old. For different types of work, the incidence of psychological abnormalities was found among those who worked in oil extraction, oil transfer, heat injection, and drilling. For different working years, the incidence of psychological abnormalities was the highest among those aged 10–20 years, followed by the age group ≤10 years. According to the linear trend test, the detection rate of different professional titles increased with the improvement of professional titles, but this trend was mainly reflected in the change of the junior and intermediate groups, and the change was not significant after the intermediate level. The incidence of psychological abnormalities among those in the shift group was higher than that of the fixed day shift group. In respect to marital status, the unmarried group had the lowest incidence of psychological abnormalities, while the divorced or widowed group had the highest incidence (Table 3).

### 3.2. Risk Scores of Major Risk Factors for Mental Health of Different Sexes

The major risk factors among males, in descending order, included occupational stress, type of work (drilling), effort–reward imbalance, shift work, and ethnicity. The risk scores for the level of occupational stress among females were higher than the risk scores for effort–reward imbalance (Table 4).

### 3.3. Comparison of Mental Health Scores between Different Genotypes of MAOA

The somatization score of the *MAOA* gene rs6323 was significantly different among different genotypes (*p* < 0.05), and it could be considered that somatization scores were different among different genotypes of rs6323. According to the average rank, TT genotype had the highest score, followed by GT genotype, while GG genotype had the lowest score.

No statistically significant difference was found in respect to the mental health score and the score of each factor among different genotypes at rs1137070 of the *MAOA* gene (*p* > 0.05), and it could not be considered that the scores of mental health, somatization, obsessive-compulsive symptoms, interpersonal sensitivity, depression, anxiety, hostility, fear, paranoia, and psychosis were different among different genotypes of rs1137070 (Table 5).

### 3.4. Comparison of DNA Methylation of the MAOA Gene between the Normal Group and Abnormal Group

The mean methylation level of all CpG sites on the fragment was calculated as the methylation level of the fragment. The methylation level of *MAOA* gene DNA methylation at different sites was different among the groups with different mental health conditions, and the methylation level of the abnormal group was lower than that of the normal group (*p* < 0.05) (Table 6).

## 4. Discussion

With the transformation of the medical model, mental health has received increasingly more attention. One study showed [42] that psychological problems are highly correlated risk factors for sleep disorders. Stress, anxiety, depression, and other negative emotions reduce the working ability of the occupational population, causing absenteeism, job burnout, and other phenomena, and seriously affects the quality of life of the occupational population [4,43,44]. The causes of mental health problems are various, and include the environment, genes, and heredity [5,6].

The results of this study found that there were differences in the incidence of psychological disorders among different ethnic groups, ages, types of work, working years, professional titles, shift work, and marital status. The incidence of psychological abnormalities was the highest in the group aged between 30 and 45 years. The reason for this finding may be that most people in this age group are married, they need to take care of children and the elderly in their lives, and they are also experiencing a period of career progress in their work, with substantial work pressure [45]. The incidence of psychological abnormalities was higher among oil extraction workers than those who engage in other types of work. Due to the special nature of oil production and poor work planning, it is difficult to carry out the work in an efficient and orderly manner. At the same time, because oil extraction work is carried out in the field, the natural environment is harsh, the living conditions are difficult, and there is less communication with the outside world, so there is no means of alleviating work-related pressures [46]. The incidence of psychological abnormalities was highest among workers with 10 to 20 years of working experience. At this stage, most workers were in important positions or leadership positions, and they assumed more responsibilities and obligations, so there were additional factors that caused mental health problems. With the increase in working years, people have to confront various sources of tension arising from their work and personal life, so negative emotions can easily contribute to the occurrence of mental health problems [47]. The incidence of psychological abnormalities rises in line with an increase in the status of professional titles. People with high ranking professional titles tend to experience a higher degree of stress and psychological problems [48]. The incidence of mental abnormalities among people whose job requires shift work is significantly higher than that observed among people who work a fixed day shift. Multiple studies have found that shift work can affect sleep, work ability and the mental health status of those in the workforce [49,50]. The results of a meta-analysis on shift work and mental health [51] showed that shift work is closely associated with an overall increased risk of adverse mental health outcomes (ES = 1.28; 95% CI = 1.02, 1.62), especially for depressive symptoms (ES = 1.33; 95% CI = 1.02, 1.74). Unmarried people have the lowest incidence of psychological abnormalities, while divorced or widowed people have the highest incidence of psychological abnormalities. Unmarried people do not need to raise children or take care of life chores, they live independently, and experience less financial pressure, so they tend to maintain a good mood and a better mental health status than married people. People who are divorced or widowed may be more likely to experience mental health problems as a result of a failed marriage or the pain of losing a loved one. Domestic scholars Liao Siqi et al. [52] also believed that marital status had a significant impact on tendencies for depression after analyzing the tendency to develop depression among 1837 oil workers in the Panjin area, and divorced oil workers scored the highest for depression tendency.

Large-scale epidemiological surveys in various countries around the world [53,54] showed that the odds ratio (OR) of residents of different races and from different regions varied greatly. When different risk factors co-existed, the risk of psychological abnormalities significantly increased. The results of this study found that different sexes had different risk factors for mental health: The risk factors of male mental health included age >45-years-old (relative to ≤30-years-old), shift work, moderate, and high occupational stress, high effort/low return, while the risk factors of female mental health included age >45-years-old (relative to ≤30-years-old), high occupational stress, and high effort/low return. In respect to shift work as a risk factor for the mental health of men working in desert oil, the risk scores of different age groups were 1.214, 1.172, and 1.369, respectively. If the shift could be changed to a fixed day shift, the risk scores could be reduced to 0.560, 0.623, and 0.512, respectively. Nascimento et al. [55] recently conducted a study of 231 nurses who worked shift work and found that shift work can result in negative emotions and lead to sleep problems. Highly intense work and intense competition can easily lead to stress. Many studies have shown that stress is an important factor affecting mental health [48,56]. As a risk factor of mental health, the risk score of occupational stress among different age groups reached 5.081, 4.767, and 3.278 for males and 3.252, 2.801, and 2.793 for females, respectively. If the source of stress can be reduced, the degree of occupational stress can be reduced to moderate stress. The risk scores of men can be reduced to 4.712, 3.916, and 2.724, respectively; the risk scores of women can be reduced to 2.972, 2.654, and 2.778, respectively. If the stress level can be reduced to low stress, the risk scores of men can be reduced to 2.840, 2.844, and 2.123, respectively, and the risk scores of women can be reduced to 1.783, 1.730, and 2.081, respectively. Foreign studies reported [57] that occupational stress can, to a certain extent, contribute to the development of psychological diseases and cardiovascular diseases. Occupational stress can affect the mental health of oil workers, generating or exacerbating anxiety and depression [58,59,60]. The risk scores of ERI among different age groups were 1.353, 1.274, and 1.264 for males, 1.538, 1.496, and 1.694 for females, respectively. Desert oil workers do not receive a corresponding return for their work efforts, which may lead to job burnout and a feeling of dissatisfaction with life, as well as anxiety, depression, and other psychological conditions. If the efforts and returns can be balanced, the risk of psychological abnormalities can be greatly reduced. Shift work, occupational stress, and pay return are all contributing factors. If these contributing factors are not addressed, the situation is bound to diminish the working ability and efficiency of employees and is more likely to result in workplace accidents [48].

In this study, it was found that the somatization factor score of TT genotype was higher than that of the GG group in regard to different psychological health groups of desert petroleum workers with rs6323 locus of the *MAOA* gene. The rs1137070 was not found to be associated with mental health problems. Aslund et al. [61] found a significant association between rs6323 polymorphism of the *MAOA* gene and anger-related traits in a controlled study on suicidal tendencies. Cao Cong et al. [62] concluded that the polymorphism of the *MAOA* gene T941G and peer aggression interact with adolescent male depression, and the genetic variation of the *MAOA* gene may affect the antidepressant treatment effect of depressed patients [63,64]. Tadic et al. [65] believed that the *MAOA* gene was related to anxiety disorder, and *MAOA* polymorphism was related to female anxiety disorder, which was not found in male patients. This study compared the methylation level of *MAOA* gene DNA methylation at different sites in the group with participants with psychological problems and found that the methylation level of the group with psychological abnormalities was lower than that of the group with psychological normality. DNA methylation influences gene expression by interacting with transcription factors or changing the chromatin structure, regulates biological genetic information at the epigenetic level, and plays an important role in growth and development [66]. Studies suggest that mental illness is caused by the interaction between genes and the environment, and DNA methylation is closely related to the occurrence of mental illness [67,68]. Domschke et al. [69] demonstrated that panic disorder is related to the hypomethylation level of the *MAOA* gene, which is consistent with the results of this study. In their study on CBT treatment of panic disorder, Ziegler et al. [70] found that the methylation level of the *MAOA* gene in patients who participated in the treatment increased after six weeks of treatment, suggesting that there was a negative correlation between the severity of panic disorder symptoms and the methylation level of the *MAOA* gene. Teroganova et al. [71] also found that methylation levels in male schizophrenia patients were lower than those in the control group, but no difference was found in women.

## 5. Conclusions

The incidence of mental health problems among oil workers differed according to age, ethnicity, type of work, working years, professional title, shift work, and marital status. After assessing the mental health risk factors of oil workers, it was concluded that shift work, occupational stress, and high effort/low return would all affect mental health. Because these factors are all contributing factors, and in order to improve the mental health of oil workers, the contributing factors should be improved first in any intervention to enhance mental health. The organizational structure can reduce workers’ occupational tension and strengthen their psychological quality by creating a positive occupational climate. The methylation level of *MAOA* gene DNA methylation at different sites was lower in the group with psychological abnormalities than in the group with psychological normalcy. It was preliminarily believed that methylation status would affect mental health, suggesting that methylation level might be used as a predictor of mental health status. The limitations and expectations of this study included the following: The index of measuring mental health was too singular, and a comprehensive index should be employed to evaluate the mental health of the occupational population in the future. Due to the limitation of expenditure and workload, only some genetic indicators of the population were tested. This study was a cross-sectional study, so no causal relationship can be inferred. A cohort study should be carried out in future studies to further clarify the relationship, and multiple site tests should be conducted to demonstrate the association between the *MAOA* gene and mental health.

## Figures and Tables

**Table 1 ijerph-17-00149-t001:** PCR primer sequences.

Primer	Direction	Sequence
MAOA rs1137070	Forward	5′-CCATTTCTCTGCCCCTCACTCA-3′
	Reverse	5′-GCATGGAGACCCCTGGGATAGT-3′
MAOA rs6323	Forward	5′-CGACCTTGACTGCCAAGATTCA-3′
	Reverse	5′-TGGCCAAGGATATGAGGAAATTGA-3′

**Table 2 ijerph-17-00149-t002:** Primer sequences

Primer	Sequence
MAOA_06F	GGGATTTGGGTAGTTGTGTTTT
MAOA_06R	AAAACATAAACACAAACRCCTCAAC
MAOA_16F	TTTTTGATATTYGGGGGGAGT
MAOA_16R	CTACACCCAATAATCCTTTCCAACTAC

**Table 3 ijerph-17-00149-t003:** Comparison of the incidence of psychological abnormalities according to different demographic characteristics (%).

Variables	Number	Number of Psychological Abnormalities	Incidence (%)	χ^2^	*p*
Sex					
Male	2208	446	20.2	1.689	0.195
Female	1423	313	22.0		
Ethnicity					
Han	2622	579	22.1	7.933	0.005
Minority	1009	180	23.7		
Age group, years					
≤30	522	95	18.2	34.085	<0.001
30–45	1785	444	24.9		
>45	1324	220	16.6		
Type of work					
Drilling	573	96	16.8	11.267	0.01
Extract oil	407	97	23.8		
Oil transportation	1166	266	22.8		
Stoker hot note work	1485	300	20.2		
Working years					
≤10	1044	234	22.4	10.56	0.005
10–20	435	111	25.5		
>20	2152	414	19.2		
Educational level					
Associate’s degree or below	2703	551	20.4	1.72	0.19
Bachelor’s degree or higher	928	208	22.4		
Professional titles					
Primary	1349	247	18.3	8.963	0.011
Intermediate	758	173	22.8		
Senior	1524	339	22.2		
Shift					
Fixed day shift	1183	216	18.3	7.423	0.007
Shift	2448	543	22.2		
Monthly income					
≤3500	1397	285	20.4	0.347	0.586
>3500	2234	474	21.2		
Marital status					
Single	359	58	16.2	6.472	0.039
Married	2943	624	21.2		
Divorced or widowed	329	77	23.4		
Smoking					
Yes	1513	309	20.4	0.362	0.562
No	2118	450	21.2		

**Table 4 ijerph-17-00149-t004:** Risk scores of major risk factors according to different sex and age groups.

			≤30			30–45			45		
			OR	BP	RM	OR	BP	RM	OR	BP	RM
Male	Ethnicity	Minority	1.000	1.036	1.036	1.000	1.093	1.093	1.000	1.145	1.145
		Han	0.885		0.917	0.706		0.772	0.436		0.499
	Type of work	Drilling	1.000	2.996	2.996	1.000	3.397	3.397	1.000	3.015	3.015
		Extract oil	0.436		1.306	0.528		1.794	0.679		2.047
		Oil transportation	0.336		1.007	0.356		1.209	0.589		1.776
		Stoker hot note work	0.317		0.950	0.556		1.889	0.582		1.755
	Shift	Fixed day shift	1.000	0.560	0.560	1.000	0.623	0.623	1.000	0.512	0.512
		Shift	2.167		1.214	1.881		1.172	2.673		1.369
	Stress level	Low stress	1.000		2.840	1.000		2.844	1.000		2.123
		Moderate stress	1.659	2.840	4.712	1.377	2.844	3.916	1.283	2.123	2.724
		High stress	1.789		5.081	1.676		4.767	1.544		3.278
	ERI index	Low effort-High return	1.000	0.670	0.670	1.000	0.614	0.614	1.000	0.552	0.552
		High effort-low return	2.020		1.353	2.075		1.274	2.289		1.264
Female	Stress level	Low stress	1.000	1.783	1.783	1.000	1.730	1.730	1.000	2.081	2.081
		Moderate stress	1.667		2.972	1.534		2.654	1.335		2.778
		High stress	1.824		3.252	1.619		2.801	1.342		2.793
	ERI index	Low effort-High return	1.000	0.779	0.779	1.000	0.599	0.599	1.000	0.440	0.440
		High effort-low return	1.974		1.538	2.498		1.496	3.849		1.694

**Table 5 ijerph-17-00149-t005:** Comparison of mental health among different genotypes of MAOA.

	RS6323			RS1137070		
	TT	GT	GG	CC	CT	TT
SCL-90 total score	162.60 ± 59.05	161.78 ± 59.57	152.94 ± 53.82	162.33 ± 57.47	157.00 ± 58.24	159.68 ± 58.16
Somatization	22.97 ± 8.93 *	22.37 ± 8.48	20.82 ± 7.85	22.39 ± 8.48	22.12 ± 8.31	21.91 ± 8.80
Forced symptoms	17.57 ± 7.18	17.60 ± 7.34	16.10 ± 6.42	17.34 ± 6.95	16.99 ± 7.16	17.19 ± 7.08
Interpersonal sensitivity	16.64 ± 6.15	16.56 ± 6.41	15.61 ± 560	16.57 ± 6.27	16.11 ± 6.27	16.33 ± 6.09
Depression	25.19 ± 9.03	25.07 ± 8.89	23.91 ± 7.72	25.31 ± 8.60	24.29 ± 8.47	24.76 ± 8.85
Anxiety	18.33 ± 6.96	17.76 ± 6.59	17.15 ± 6.21	18.052 ± 6.64	17.34 ± 6.57	18.05 ± 6.78
Hostile	10.17 ± 4.31	9.95 ± 4.06	9.55 ± 3.87	10.04 ± 4.08	9.73 ± 4.07	10.02 ± 4.23
Terrorist	12.30 ± 5.077	12.18 ± 4.93	11.73 ± 4.90	12.42 ± 5.03	11.86 ± 4.88	12.05 ± 5.03
Paranoid	11.17 ± 4.11	11.08 ± 4.03	10.55 ± 3.82	11.04 ± 3.97	10.84 ± 3.91	11.02 ± 4.11
Psychotic	17.77 ± 6.64	17.80 ± 6.85	16.96 ± 6.29	17.83 ± 6.63	17.09 ± 6.55	17.62 ± 6.61

Note: Comparison of mental health scores among genotypes at different sites of MAOA, * *p* < 0.05.

**Table 6 ijerph-17-00149-t006:** Comparison of DNA methylation of the *MAOA* gene among different mental health groups.

*MAOA* Gene	Psychological Normal	Psychological Abnormal	*T*	*p*
CpG1	0.33 ± 0.17	0.11 ± 0.18	4.309	< 0.001
CpG2	0.33 ± 0.17	0.11 ± 0.18	4.309	< 0.001
CpG3	0.20 ± 0.10	0.07 ± 0.12	4.042	< 0.001
CpG4	0.31 ± 0.16	0.11 ± 0.15	4.417	< 0.001
CpG5	0.31 ± 0.16	0.09 ± 0.16	4.712	< 0.001
CpG6	0.23 ± 0.14	0.08 ± 0.14	3.672	< 0.001
CpG7	0.32 ± 0.17	0.10 ± 0.17	4.435	< 0.001
CpG8	0.30 ± 0.16	0.10±0.17	4.155	< 0.001
CpG9	0.14 ± 0.08	0.06 ± 0.10	3.035	< 0.001
CpG10	0.31 ± 0.16	0.10 ± 0.16	4.498	< 0.001
CpG11	0.31 ± 0.16	0.09 ± 0.16	4.712	< 0.001
CpG12	0.19 ± 0.11	0.06±0.10	4.234	< 0.001
CpG13	0.32 ± 0.17	0.11±0.18	4.113	< 0.001
CpG14	0.33 ± 0.17	0.11 ± 0.18	4.309	< 0.001
CpG15	0.24 ± 0.13	0.08 ± 0.14	4.062	< 0.001
CpG16	0.33 ± 0.17	0.10 ± 0.18	4.505	< 0.001
CpG17	0.33 ± 0.17	0.10 ± 0.18	4.505	< 0.001
CpG18	0.26 ± 0.14	0.08 ± 0.13	4.563	< 0.001
CpG19	0.34 ± 0.18	0.11 ± 0.19	4.262	< 0.001
CpG20	0.44 ± 0.22	0.14 ± 0.23	4.571	< 0.001
CpG21	0.21 ± 0.11	0.08 ± 0.12	3.874	< 0.001
CpG22	0.28 ± 0.14	0.10 ± 0.14	4.406	< 0.001
CpG23	0.44 ± 0.22	0.14 ± 0.23	4.571	< 0.001
CpG24	0.24 ± 0.13	0.09 ± 0.13	3.954	< 0.001
CpG25	0.38 ± 0.19	0.13 ± 0.20	4.395	< 0.001
CpG26	0.43 ± 0.20	0.16 ± 0.20	4.627	< 0.001
CpG27	0.30 ± 0.15	0.10 ± 0.15	4.569	< 0.001
CpG28	0.28 ± 0.14	0.10 ± 0.15	4.255	< 0.001
CpG29	0.46 ± 0.23	0.14 ± 0.24	4.668	< 0.001
CpG30	0.37 ± 0.19	0.11 ± 0.19	4.690	< 0.001
CpG31	0.43 ± 0.22	0.14 ± 0.22	4.518	< 0.001

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
