# Peer review of "Cross-Sectional Survey of Mental Health Risk Factors and Comparison of the Monoamine oxidase A Gene DNA Methylation Level in Different Mental Health Conditions among Oilfield Workers in Xinjiang, China"

_ijerph, 2019, doi:10.3390/ijerph17010149_

Round 1

Reviewer 1 Report

This is a good study, which was conducted on a big amount of participants, and have revealed interesting correlations. Moreover, the laboratory testing of altered MAOA gene DNA methylation, which emerges as a possible pathogenetically relevant epigenetic mechanism in mental disorders was also performed.

With that there are some minor questions to the authors.

Please describe the randomization procedures (for example, how did you avoid the Order Bias and other problems). (Bradburn, N.M. and Sudman, S. (2011). The Current Status of Questionnaire Research. In Measurement Errors in Surveys (eds P.P. Biemer, R.M. Groves, L.E. Lyberg, N.A. Mathiowetz and S. Sudman).) Please describe the Sample size calculations in the laboratory testing (were you measured DNA methylation). Lines 29-30 Abstract. Please rewrite and make the clear conclusion. Line 254. You wrote “studies have shown” while there is only one Ref.

Thank you

Author Response

Dear Reviewer,

Thank you for your valuable advice, which is very helpful to my research. According to your opinions, I have revised the paper. I am looking forward to your reply.

Best regards,

Xue Li

1.Please describe the randomization procedures.

Response:According to your suggestion, I have included a description of random methods in the methods section of this article (lines 128-130).

Please describe the Sample size calculations in the laboratory testing (were you measured DNA methylation).

Response:In article 2.3 The experimental method, the experimental method is described, including DNA methylation test.

Lines 29-30 Abstract. Please rewrite and make the clear conclusion.  

Response:This conclusion is redescribed.

Line 254. You wrote “studies have shown” while there is only one Ref.

Response:“Studies have shown”instead of“One Study showed ”。

Reviewer 2 Report

Tiang Jiang and coworkers present an interesting research article that links mental health and the methylation status of monoamine oxidase A (MAOA) gene. They suggest that lower methylations levels of rs6323 (R297R / Arg297Arg), a well-known SNP, where the G allele encodes for the higher activity form of the enzyme that degrades serotonin, dopamine, epinephrine, and norepinephrine are related with psychological problems in an according to this study performed in petroleum workers.  The topic is interesting, and the opportunity to link epigenetic modifications with human behavior seems to be extremely thought-provoking, no pun intended. However, the manuscript has serious issues, and it is absolutely not ready for publication at the International Journal of Environmental Research and Public Health for the following reasons:

However minor, it is surprisingly not formatted correctly for a review process, there is not numbered line or numbered pages, so the next comments would need to be taken in the context. And I will start following the order of the text instead of the importance of the issues due to this issue. Why are they trying to describe the methods in the titer? They do NOT need to mention in the titer that they performed the study in 3,631 people. This needs to be described in the methodology. Acronyms (MOAO in this case) should be avoided in the titer. In the introduction, there is not a clear link between MOAO as an enzyme and how methylation could be implicated in the catalytic activity of the enzyme by the epigenetic control of the gene. In general, they mainly focus on the psychological aspect and the molecular part of the research is surprisingly very weak. They never present how the DNA methylation analysis was performed! They have a section in material and methods that stands for “Laboratory quality control”. That is awkwardly written. “The laboratory operation was carried out in strict accordance with the requirements of the laboratory (sic.)” what does this mean? Which one are these “requirements”? is it a standard BSL2? Why is this important??

Followed by ”White coats, disposable gloves, and masks were worn, and blood collection was performed by the hospital nurses between 9 am and 11 am. Quality control was carried out throughout the whole process, before, during and after sample analysis, and was done so

in strict accordance with the quality control standards established by the laboratory.”   I have never seen anything like that, what a waste of space for information that is not need it. And the real methods of how they get they results are just MISSING!

“The samples were stored in the refrigerator at -80°C, and to prevent repeated dissolution.” Dissolution of what, how do they process the samples, which materials they use to separate the different cell components of the blood, how do they isolate the DNA, how the METHYLATION analysis was performed????  

“During the experiment, the instructions were strictly adhered to. Contaminated areas were clearly distinguished from clean areas to avoid false positive results caused by cross contamination.” Incomplete sentences… which instructions of what????

“An ultraviolet lamp was used before polymerase chain reaction (PCR), and the disposable

nozzle, Ep tube, PCR tube, and other consumables were sterilized using an autoclave prior to

use. A blind method was adopted for genetic testing, and samples were randomly selected at

10% to check the consistency of the results. When the results of the retest were inconsistent

with the initial test, the experiment was repeated to check again”  

 They say they performed PCR, no primers, reagents of protocols are described, most importantly, how the METHYLATION analysis was performed?? This is the heart of the article. And nothing was described in the methods of how they get their results.

Minor issues:

Issues with formatting, double-space between lines, no space between period, or capital letters after semicolons and different fonts in some parts of the text, including tables. In the introduction (second page, 3rd line) authors need to be consistent with how they present the data, they start talking on percentages and suddenly they jump to … 173 million people in China…. after that they are back to … 12.1% in West Africa… They mention … “Many studies” and they cite two.

This manuscript has serious issues, and it is absolutely not recommended for publication at the International Journal of Environmental Research and Public Health.

Author Response

Dear Reviewer,

First of all, I don't know which link caused the problem of the unnumbered line of the article you received, which may increase your workload, for which I am very sorry.Thank you for your valuable advice, which is very helpful to my research. According to your opinions, I have revised the paper, 2.3 The experimental method was added to the article.Other questions you asked have also been modified.I am looking forward to your reply.

Best regards,

Xue Li

They do NOT need to mention in the title that they performed the study in 3,631 people. Acronyms (MOAO in this case) should be avoided in the title.

Response:Change the title of the article to“Cross-Sectional Survey on the Evaluation of mental health risk factors and comparison of Monoamine Oxidase A gene DNA methylation level in different mental health conditions of Oilfield Workers in Xinjiang, China”.

In the introduction, there is not a clear link between MOAO as an enzyme and how methylation could be implicated in the catalytic activity of the enzyme by the epigenetic control of the gene.

Response:Add a description of DNA methylation and the MAOA gene in the introduction (lines 85-97).

How do they isolate the DNA, how the METHYLATION analysis was performed?

Response:The article "2.3. The experimental method" described the experimental method in detail, and modified the content of the “Quality control” .

Issues with formatting, double-space between lines, no space between period, or capital letters after semicolons and different fonts in some parts of the text, including tables.

Response:These issues were carefully checked and revised in the article.

In the introduction (second page, 3rd line) authors need to be consistent with how they present the data, they start talking on percentages and suddenly they jump to … 173 million people in China…. after that they are back to … 12.1% in West Africa… They mention … “Many studies” and they cite two.

Response:I have also modified these issues you mentioned in the article.(44-45 lines)(72lines)

Round 2

Reviewer 2 Report

Ting Jiang et al. present a significantly improved version of the manuscript that is now suitable for publication.